# Antenna surface plasmon emission by inelastic tunneling

Cheng Zhang[1], Jean-Paul Hugonin[1], Anne-Lise Coutrot[1], Christophe Sauvan [1], François Marquier [2] & Jean-Jacques Greffet [1]*

Surface plasmons polaritons are mixed electronic and electromagnetic waves. They have become a workhorse of nanophotonics because plasmonic modes can be confined in space at the nanometer scale and in time at the 10 fs scale. However, in practice, plasmonic modes are often excited using diffraction-limited beams. In order to take full advantage of their potential for sensing and information technology, it is necessary to develop a microscale ultrafast electrical source of surface plasmons. Here, we report the design, fabrication and characterization of nanoantennas to emit surface plasmons by inelastic electron tunneling. The antenna controls the emission spectrum, the emission polarization, and enhances the emission efficiency by more than three orders of magnitude. We introduce a theoretical model of the antenna in good agreement with the results.

[1] Laboratoire Charles Fabry, Institut d'Optique Graduate School, CNRS, Université Paris-Saclay, 91127 Palaiseau, France. [2] Laboratoire Aimé Cotton, Ecole Normale Supérieure de Paris-Saclay, CNRS, Université Paris-Saclay, 91405 Orsay, France. *email: jean-jacques.greffet@institutoptique.fr

Surface plasmon polaritons (SPPs) are mixed electronic and electromagnetic modes propagating along the interface between a metal and a dielectric[1]. The corresponding electromagnetic field can be confined to nanometer scale and has a decay time on the order of 10 fs so that a new generation of devices can be designed[2–5]. SPPs are widely used for sensing and also for information technology[6–11]. An important limitation of surface plasmons is that in most cases, they are optically excited by a diffraction-limited beam so that the device size is larger than the wavelength. In order to fully benefit from the field confinement, it is necessary to generate SPPs in the near field with a subwavelength device. Electrical generation of surface plasmons has been demonstrated using LED-based platforms[12–16] with time scales on the order of 1 ns. Ultrafast emission can be achieved using light emission by inelastic tunneling (LEIT) through a tunnel junction.

LEIT through a planar tunnel junction was discovered by Lambe and McCarthy[17]. This process is rather inefficient with a typical efficiency on the order of one photon per a million electrons. However, LEIT has important advantages. Firstly, it is intrinsically fast. The fundamental limit is given by the tunneling time[18,19], which is on the order of $h/eV$ where $e$ is the electron charge, $V$ the applied voltage on the order of 1 V and $h$ is Planck's constant yielding a time limit on the order of 4 fs. In practice, the limit is given by the circuit time constant RC. Photon sources based on LEIT have been operated at 1 GHz[20]. Secondly, LEIT can be a highly localized source. The source current density can be confined in a nanoantenna or in a metallic tip enabling electromagnetic excitation localized at the nanometer scale. Both photon[21] and plasmon[22–24] emission have been reported using scanning tunneling microscope (STM) tips. More recently, surface plasmon emission has been reported with metallic microstructures[25] and molecular junctions[26]. Theoretical models are available for LEIT[18,27–31], and for the role of the gap plasmon mode[32]. Hence, antenna surface plasmon emission by inelastic tunneling (ASPEIT) appears to be a suitable candidate for ultrafast and highly localized plasmon emission[33].

However, several issues need to be solved to use inelastic tunneling as a light source. It is required to control the emission spectrum, the angular emission pattern and to increase the emission efficiency. In order to tackle these issues, resonant nanoantennas can be used. The rationale for using an antenna is based on different properties. Firstly, a resonant antenna can select the frequency emission[34–36]. Secondly, a resonant antenna can be designed in order to control both the polarization and the angular emission pattern[37,38]. Thirdly, a resonant plasmonic antenna contributes to the local density of states in the junction. If the contribution of the antenna mode is larger than the contribution of the non-radiative modes, it becomes possible to avoid quenching and therefore increase the radiative efficiency[35,39,40] in a controlled and deterministic way. It is important to stress that there is no fundamental limit to the LEIT efficiency. We note in particular that in the microwave regime, the non-radiative modes can be suppressed using dissipation-less superconductors. Furthermore, impedance matching between the junction and a 50-Ω line can be achieved by microwave engineering so that photon emission in a 50-Ω line can reach very high efficiencies[28]. Photon emission in the visible by electrically driven optical antennas has been reported demonstrating a spectral control of the emission[41,42] and also a directional control[43]. Spectral control of LEIT using resonant nanocubes has been demonstrated recently[44] although only a small fraction of the tunneling current was affected by the antenna. These results demonstrate the potential of plasmonic nanoantennas to tailor LEIT. All these demonstrations deal with light emission. No antennas have been reported to emit surface plasmons so far. Reported surface plasmon sources

are based on the use of a metallic tip[22–24] or a planar junction[25,26]. While it is known that a localized gap plasmon takes place between the tip and the surface, no control of this mode using antennas has been reported.

There are still many issues to be solved before developing electrical surface plasmon sources using inelastic tunneling assisted by antennas: (i) the tunnel barrier width, which plays a critical role in the success of the emission process, is not deterministically controlled in the reported experiments, (ii) the lifetime of the antenna is still an outstanding issue, (iii) LEIT is limited to very small power on the order of 1 fW for STM junctions or on the order of 10 fW for planar junctions while pW are needed for practical applications, (iv) the coupling between the antenna mode and the propagating plasmon mode has to be optimized, (v) a theoretical analysis of the emitted power as a function of the current fluctuations in the presence of an antenna is needed in order to analyze and optimize the emission.

Here, we report the design, fabrication and characterization of antennas to emit surface plasmons propagating along an aluminum/air interface. We also report an original theoretical model introducing a figure of merit of the antenna showing that the relevant figure of merit is not the Purcell factor but the spatially averaged enhancement factor in the junction barrier. The antenna produces a narrow emission spectrum and provides an enhancement of the efficiency as compared to a planar junction larger than three orders of magnitude. The emitted power is on the order of 10 pW, four orders of magnitude larger than an STM tip.

## Results

**ASPEIT junction and electrical characterization.** The electrical surface plasmon source consists in a gold nanopatch antenna deposited on an Al/AlOx interface (see Fig. 1a). A plasmon is emitted when an electron tunnels inelastically from gold to aluminum through a 3-nm-thick AlOx tunnel barrier as shown in Fig. 1b. The gap between gold and aluminum supports gap plasmons, which are reflected at both edges of the antenna separated by a width $D = 128$ nm forming a Fabry–Perot cavity[45] (Fig. 1a). Due to partial transmission at the edges, plasmons are emitted by the antenna and propagate along the Al/air interface. In order to observe them, the aluminum thickness was chosen to be 25 nm so that plasmons can leak in the glass substrate. The geometry of the antenna needs to be optimized so that the field of the antenna mode is efficiently coupled to both the current density and the radiated plasmonic field[34]. The thin gap and the resonant behavior of the Fabry–Perot cavity serves to enhance the field in the plasmonic cavity in order to couple to the current density. The radiation of propagating surface plasmons depends on the height of the antenna. Both effects are captured by the theoretical form of the signal derived in the Supplementary Note 5 where we introduce a figure of merit, which is the spatially averaged enhancement factor. In order to increase the emitted optical power, we have fabricated a finite size array of 25 μm width where the 60-μm-long and 128-nm-wide antennas are periodically arranged with a period of 400 nm (Fig. 1c). We checked that the device is operating in the tunnel regime by measuring the current density ($J$) as a function of the applied voltage. The characteristic $J(V)$ curve is depicted in Fig. 2 in linear plot (Fig. 2a) and semi-log plot (Fig. 2b) displaying the typical exponential dependence. A fit of the data with Simmon's model[46] allows the estimation of the barrier thickness (3 nm) and the mean barrier height (2.01 ± 0.03 eV) using an effective mass of the electron of $0.23m_e$[47]. As an additional check of the tunnel regime, we show in the Supplementary Note 4 that the current is proportional to the area of the device for four different devices and we also show that the $J(V)$ curve does not depend on temperature.

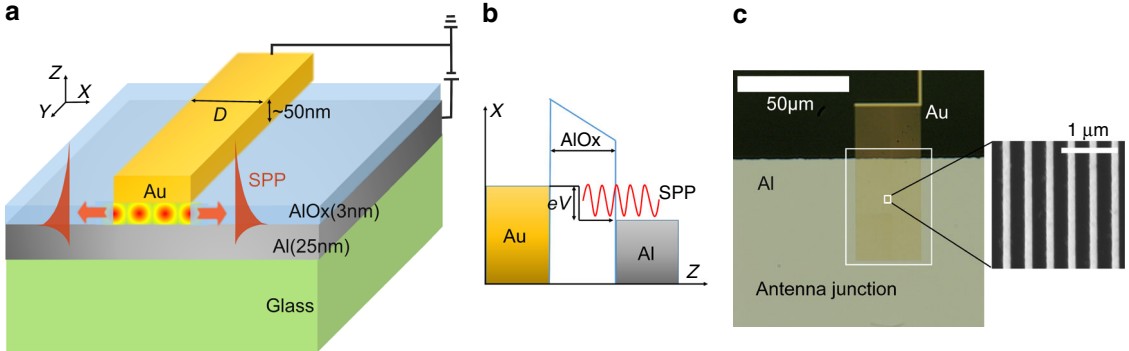

**Fig. 1** Principle of antenna surface plasmon emission by inelastic tunneling (ASPEIT). **a** Schematic of the device consisting of a gold patch antenna (thickness 50 nm and width $D$) on an aluminum film (thickness 25 nm) with a 3-nm-thick tunnel barrier of AlOx in between. Gap plasmons propagating along the x-axis under the gold patch are reflected at the edges forming a Fabry–Perot cavity. The surface plasmons transmitted at the edges propagate along the Al/air interface. They are observed by leakage radiation microscopy through a finite thickness of Al. When applying a voltage bias V, SPPs can be excited by inelastic tunneling. **b** Energy level diagram for the metal–insulator–metal (MIM) tunnel junction with a voltage bias V. The black arrow shows the maximum electron energy loss, which is the maximum plasmon energy. **c** Optical microscope image showing the periodical antenna junction (inside the white square). A SEM image shows a zoom of the array of linear antennas (width $D = 128$ nm, period 400 nm)

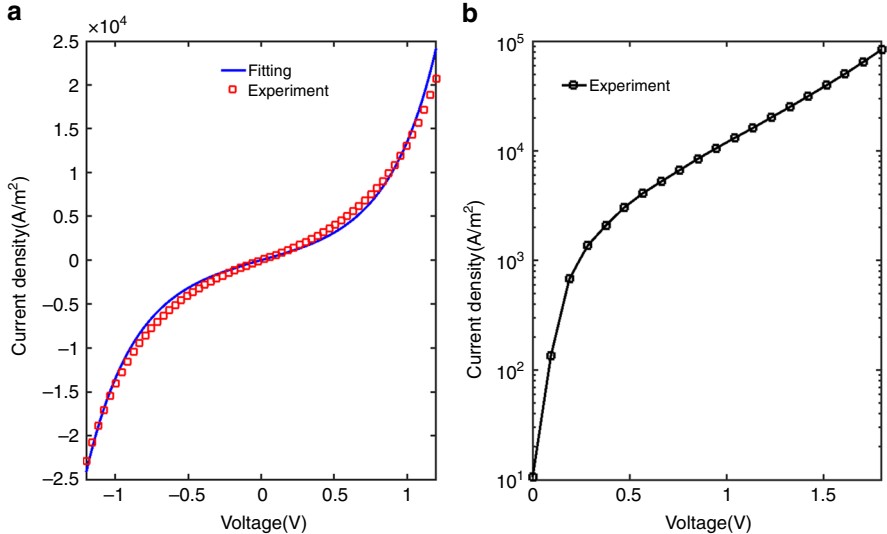

**Fig. 2** $J(V)$ characteristic of ASPEIT junction. **a** Red dots: $J(V)$ experimental data, blue line: fitted curve based on the Simmons model (fitting parameters: barrier thickness of 3 nm and mean barrier height of 2.01 ± 0.03 eV, effective electron mass $0.23m_e$). **b** Semi-log plot of $J(V)$ showing the exponential behavior

**Antenna surface plasmon emission**. We now turn to the optical characterization of the emitted surface plasmons. The optical observation of the SPPs emitted by the Al/AlOx/Au junction is based on leakage radiation microscopy (LRM) using an inverted optical microscope. We use an oil objective (NA = 1.3) to collect the leakage of the emitted SPPs through the glass substrate when the junctions are biased. We show in Fig. 3a an optical microscope image of the device taken through the glass substrate. The bright area in the center corresponds to the array of antennas with a width of 25 μm. The direction of the antennas is along the y-axis. It is seen that on both sides of the array of antennas, light is also collected. This is due to radiative leakage of surface plasmons propagating away from the antennas. To prove the plasmonic character of the emitted light, we averaged the intensity along y in the area indicated by the dashed rectangle shown in Fig. 3a. The result is plotted as a function of x in Fig. 3b in a semi-log scale showing clearly an exponential decay with a decay length of 4.9 μm, which is smaller than theoretically predicted (8.5 μm at 850 nm) when using the data of ref. [48]. This value is known to be highly dependent on the deposition procedure[48]. To further

check the plasmonic character of the emitted light, we recorded the far-field angular spectrum in the back focal plane as displayed in Fig. 3c. It is clearly seen that light is predominantly emitted for wavevectors slightly larger than the vacuum wavevector $k_0 = \omega/c$, a clear signature of surface plasmons propagating along an Al/air interface. An interesting non-uniform intensity pattern due to the antenna periodicity is observed as a function of the angle. To understand the data, we have developed a theoretical model of ASPEIT. A detailed derivation is given in Supplementary Note 5. The simulated emission pattern at 850 nm (1.46 eV) shown in Fig. 3d is in good agreement with the experimental data, which have a broader spectrum.

We now study the spectrum of the emitted field. To proceed, we collect the signal emitted toward the substrate and send it to a spectrometer (Andor Shamrock 750i). The data are shown in Fig. 4a–c and the theoretical simulations are shown in Fig. 4 (d–f) for three different bias voltages 1.4, 1.5 and 1.6 V. Note that the calculations account for the detector responsivity, responsible for the decay of the signal at low energy. We observe two spectral peaks at 1.2 and 1.5 eV that are well recovered by the model.

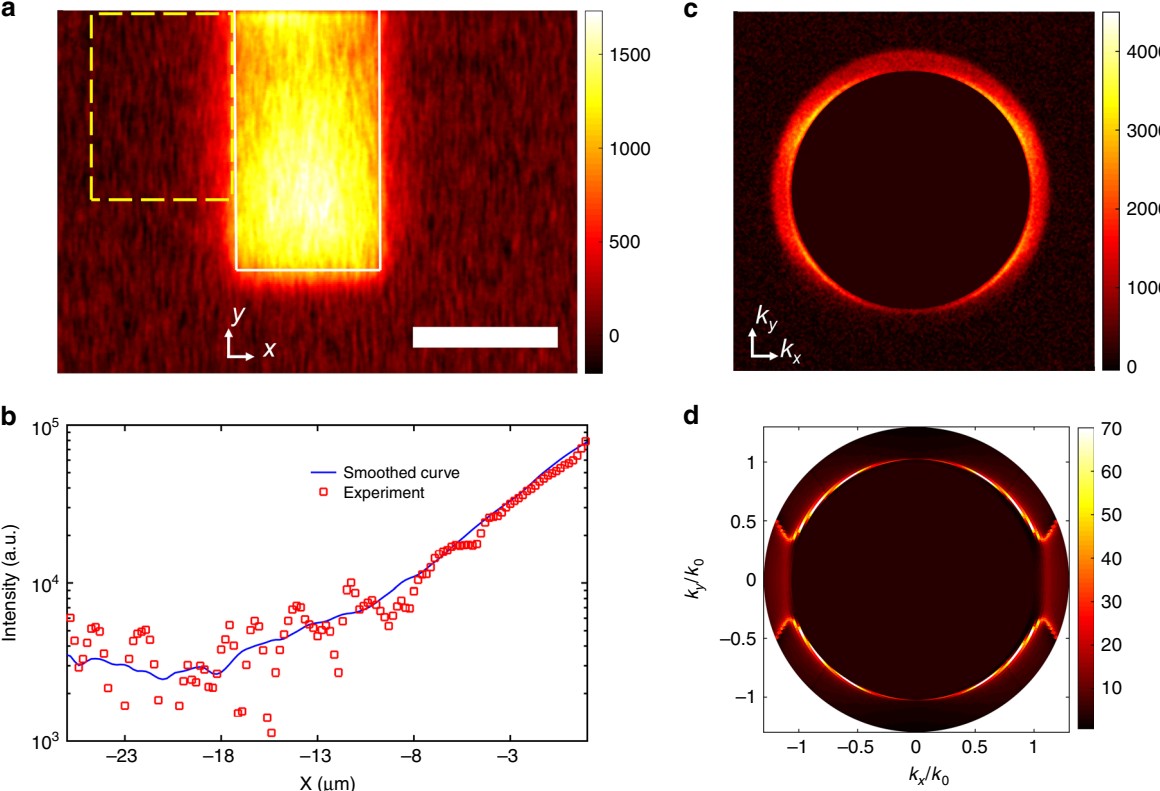

**Fig. 3** Plasmon emission. **a** Image of the source plane showing the electrical excitation of SPP (under a biased voltage of 1.6 V). The white rectangle indicates the area of the array of antennas (the scale bar is 25 µm). **b** Cross-section profile along the x-axis in a semi-log plot is averaged over the y-axis in the yellow dashed line rectangle (red dots are experiment data, the blue smoothed curve is a guide to the eye to see the SPP profile, the propagation length of the SPP is 4.9 ± 0.3 µm. **c** Experimental back focal plane image, showing the plasmon emission between $k/k_0 = 1.0$ (light cone in air) and $k/k_0 = 1.3$ (immersion objective numerical aperture), where $k_0 = \omega/c$ and $k^2 = k_x^2 + k_y^2$, $k_x = n k_0 \sin\theta \cos\phi$, where $n = 1.5$ is the refractive index of the substrate. **d** Simulated back focal plane image (at wavelength of 850 nm and the width of patch is 132 nm)

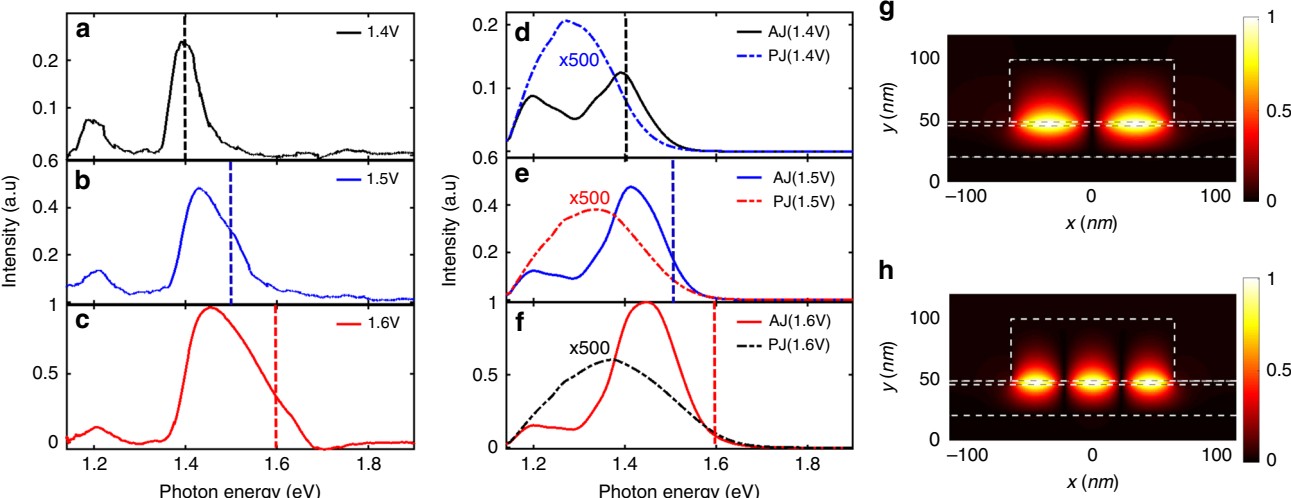

**Fig. 4** Controlling the emission spectrum. **a–c** Experimental SPP emission spectra at three different biases (1.4–1.6 V, width of patch is around 128 nm, period of patch is 400 nm, spectra intensity have been normalized by the maximum power with the bias of 1.6 V); **d–f** theoretical normalized SPP emission spectra with same bias voltage as the figures (**a–c**). The emission spectrum shows two spectral peaks due to the excitation of two resonant modes of the antenna. The peaks are broadened due to the inhomogeneity of the antenna width. The simulations result from an average over seven different widths varying from 116 to 140 nm to account for width fluctuations along the wires in the experiments. The dotted line in each figure shows the corresponding normalized spectrum from a planar junction with the same voltage bias (intensity of the spectra has been multiplied by a factor of 500). All the theoretical emission spectra are corrected by the quantum efficiency of the spectrometer responsible for the decay at low energies. **g, h** Theoretical magnetic field distribution[54] ($H_y$ in the X–Z plane) of the two gap modes of a 132-nm-wide antenna corresponding to the lower photon energy at 1.2 eV (**g**) and the higher photon energy at 1.5 eV (**h**)

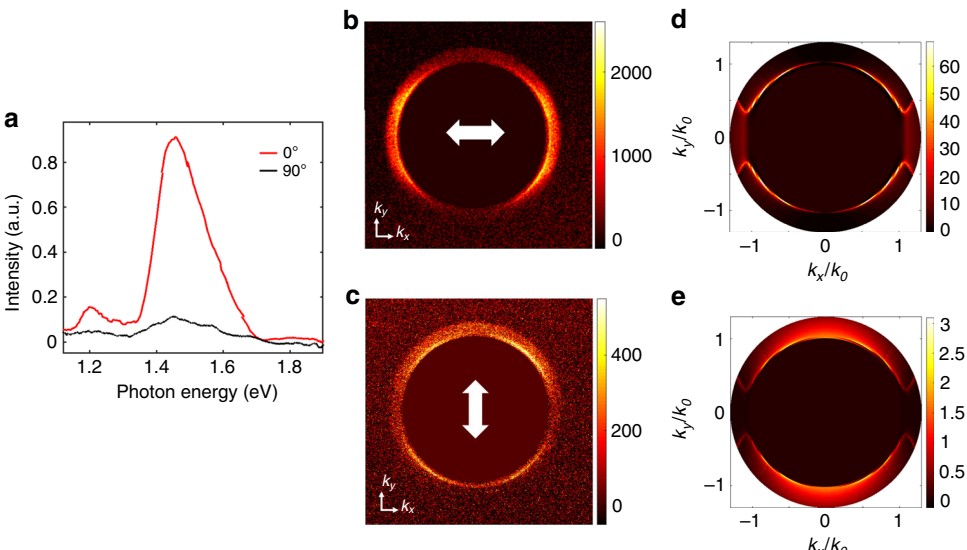

**Fig. 5** Polarization-dependence of the ASPEIT junction. **a** SPP emission spectra with two different polarized directions (0° stands for the direction, which is perpendicular to the long wire patch antenna, while 90° stands for the direction that is parallel to the wire.) **b**, **c** Back focal plane image, showing the plasmon emission between $k/k_0 = 1.0$ and $k/k_0 = 1.3$, with two different polarization direction of 0° and 90° (the white arrows indicate the polarization direction). **d**, **e** Simulated back focal plane images (at 1.45 eV and $D = 132$ nm) associated with the figure (**b**, **c**)

Here, we see the ability of the antenna to control the light emission spectrum. We note that to recover the spectrum shape and width, we had to average over different antenna widths in order to account for width fluctuations of the antenna (see Supplementary Fig. 2). The two peaks correspond to the two modes of the Fabry–Perot cavity shown in Fig. 4g–h. We report in Supplementary Fig. 10 emission spectra of antennas with different widths showing different emission frequencies. A cutoff of light emission is expected around $eV/h$ (indicated by a vertical dashed line in Fig. 4a–f for inelastic light emission) but not observed. The spectrum is significantly broadened and shifted from the theoretical $eV/h$ value by the finite temperature (300 K) and by the antenna resonance. When repeating the simulation at 0 K as shown in Supplementary Fig. 9, a clear cutoff is observed. For the sake of comparison, we plot in Fig. 4d–f the theoretical emission spectrum of a planar junction at three different voltages multiplied by a factor of 500. This shows the two roles of the antenna: spectral control and efficiency enhancement.

**Polarization emission of the ASPEIT junction**. We now move to the study of the polarization emission properties of the antenna. Figure 5 shows the far-field emission of the device when using an analyzer aligned perpendicular to the antennas (0°) or parallel to the antennas (90°). It is seen that the signal drops significantly at the emission peak showing that light emission by the antennas is polarized. The far-field emission pattern confirms that this polarization behavior is due to the plasmons. Figure 5b, c shows the back focal plane image for two orientations of the analyzer.

**Emitted power and efficiency**. An important figure of merit of an electrical source of surface plasmon is the emitted power. We detected $3.6 \times 10^6$ photons per second by using ASPEIT (with a voltage bias of 1.6 V). Correcting for the collection efficiency, this corresponds to $1 \times 10^8$ photons emitted per second (~23 pW, using a photon energy of 1.45 eV). It is interesting to compare with a STM tip[22] ($10^4$ photons per second, 3 fW), with a planar junction[25] ($4.3 \times 10^5$ photons per second, 100 fW). The increased emitted power is both due to the larger size of our source (25 μm

× 45 μm) and to the electron to photon conversion efficiency enhancement.

As already mentioned, the low efficiency of the inelastic tunneling is due to the fact that the photonic local density of states in a tunnel junction is dominated by non-radiative modes. The key to increase the efficiency is therefore to design a plasmonic antenna with a resonant plasmon mode fulfilling two conditions[35]: (i) it provides a contribution to the local density of states larger than the non-radiative modes, (ii) its losses are dominated by radiative losses. We designed our antennas to fulfill these criteria. In order to measure the efficiency enhancement, we have compared the emission by the array of antennas with the emission of the same sample at a different position with no antennas (see Supplementary Note 9). We found an averaged efficiency of $6.0 \times 10^{-10}$ photons per electron. This low efficiency is due to the low transmission of the 25 nm thick Al layer in the near infrared and also to the good quality of the surface, whose roughness is lower than 1 nm. Such a low roughness is required in order to fabricate a good plasmonic cavity. We now compare with the efficiency of light emitted by the area covered with antennas shown in Fig. 3a. The optical power is integrated over the array of antennas (dashed rectangular region in Fig. 3a) and normalized by the corresponding intensity assuming that the current density is uniform. We found $1.6 \times 10^{-6}$ photons per electron indicating a 2700 efficiency enhancement compared to the planar junction. Taking into account the plasmon propagation loss, it corresponds to an electron-to-plasmon efficiency on the order of $10^{-5}$.

**Discussion**

To analyze what are the key factors playing a role in this enhancement, we have derived a theoretical model of light emission by an antenna (see Supplementary Note 5). We compute the fields emitted by a fluctuating current density in the tunneling gap. The power spectral density of the current fluctuations is given by[29]:

$$\langle I^2 \rangle(\omega) = \frac{1}{1 - \exp\left(-\frac{eV}{k_B T}\left(1 - \frac{\hbar\omega}{eV}\right)\right)} eI_0 \left(1 - \frac{\hbar\omega}{eV}\right), \quad (1)$$

where $I_0$ is the tunneling current, $e$ the electron charge, $\omega$ the

frequency, $\hbar$ the reduced Planck's constant, $k_B$ is Boltzmann's constant and $T$ is temperature. To compute the emitted field, we use the reciprocity theorem[32]. The efficiency defined as the number of emitted photons due to plasmon leakage per electron can be cast in the form (see Supplementary Note 5):

$$\eta_{e-p}(\omega_0) = \left[\frac{Z_0}{R_k}\right] \frac{\Delta\omega}{\omega_0} \frac{n}{4} \left(\frac{t}{\lambda_0}\right)^2 \overline{|K^l(\mathbf{u},\omega_0)|^2} \frac{\langle I^2\rangle(\omega_0)}{eI_0} \Delta\Omega, \quad (2)$$

where $R_k = h/e^2$ is the quantum of resistance, $Z_0 = \mu_0 c$ the vacuum impedance, $c$ light speed, $t$ the barrier thickness, $\lambda_0$ and $\omega_0$ are the central wavelength and central frequency of a resonant antenna, respectively, $\Delta\Omega$ is the related solid angle, $\Delta\omega$ is the spectral width of the emitted peak, and $\overline{|K^l(\mathbf{u},\omega_0)|}$ is the field enhancement factor (with corresponding polarization and emission direction) spatially averaged over the whole junction barrier. It is defined as the ratio of the electric field in the junction and the electric field of an incident plane wave coming from the substrate in the emission direction. The quantity $nZ_0(t/\lambda)^2\langle I^2\rangle(\omega)$ is essentially the power radiated by a dipole with length $t$ in a homogeneous medium with refractive index $n$. The role of the antenna appears in the spectrally dependent factor $|K^l(\mathbf{u},\omega_0)|^2$. For an angle, polarization and frequency corresponding to the resonant excitation of a plasmon, it takes the value $5\times10^{-3}$ for a planar junction due to the low transmission through the 25-nm-thick aluminum layer. However, it is on the order of 70 in the presence of the resonant antenna, which thus provides an enhancement of four orders of magnitude. It is seen that this factor is the key to improve the efficiency by increasing the field in the junction using the antenna.

As shown before, the theoretical model predicts correctly the emission spectrum, the angular emission pattern and the polarization. However, the photon per electron efficiency is only qualitatively predicted. The theoretical model predicts an antenna efficiency of $1.1\times10^{-7}$ (under a voltage of 1.6 V), which is one order magnitude lower than the experimental value $1.6\times10^{-6}$. We also find a difference for the antenna junction efficiency normalized by the planar junction efficiency. The experimental value is 2700 whereas the model predicts 400. We attribute this difference to the simplified model of the current fluctuation correlation function used. It has already been reported in the literature[49,50] that this model underestimates the efficiency and it has been suggested that light emission could be due to hot electrons in the metal. We have computed this contribution and we found a marginal change. We thus conclude that the origin of the discrepancy is due to the simplified correlation function of the current density. In summary, the antenna appears to be a powerful tool to enhance the efficiency of LEIT, which has been known for decades to be an inefficient light emission process. Nonetheless, the experimental electron to plasmon conversion efficiency ($\sim10^{-5}$) is still low. We now discuss how it could be improved. As shown in Supplementary Fig. 13, replacing aluminum and gold by silver allows gaining one order of magnitude in the electron to photon efficiency ($1.4\times10^{-5}$) so that the electron to plasmon efficiency could be on the order of $10^{-4}$.

In conclusion, we have reported ASPEIT. We have shown that the emission spectrum can be tuned by varying the width of the antenna. An important feature of the antenna is its ability to enhance the efficiency of the coupling between electrons and plasmons. We have demonstrated a more than three orders of magnitude enhancement of the electron to photon conversion efficiency when comparing with a planar junction. The ASPEIT operates in the 10 pW regime, four orders of magnitude above the fW regime of a STM-based junction. We have introduced a theoretical model of ASPEIT that accounts for the observed spectrum, polarization structure and is able to predict the

efficiency enhancement. This model provides guidance to further improve the efficiency. Using the model, we find that a silver/alumina/silver junction could reach an efficiency at $10^{-4}$ plasmons per electron. This raises the prospect of an efficient, ultrafast and highly localized electric surface plasmon source.

## Methods

**Devices fabrication**. The ASPEIT tunnel junctions are fabricated on a standard coverslip (VWR, 0.15 mm thick). The bottom Al electrode is fabricated by UV photolithography via a mask aligner (MJB4). An Ebeam evaporator (MEB 550, Plassys) is employed to do the Al deposition. After the lift-off, a dry thermal oxidation process in a furnace is used to generate an ultrathin AlOx layer. The Au top electrode is fabricated by using an electron-beam lithography (Nanobeam NB4) system. All the detailed fabrication processes can be found in Supplementary Note 1.

**Electrical characterization**. The electrical characterization of our junction is performed via a source meter (Agilent B2902A). The IV measurements are conducted by a Quick IV Measurement software provided by keysight technologies website. For all experiments, the Au electrodes are grounded and the Al electrodes are biased. To avoid any possible dielectric breakdowns, each IV curve is within ±1.8 V. To avoid heating the junction, we use a time period of 1.5 ms for every voltage step. When performing the electroluminescence (EL) measurement, we apply a lower voltage (<1.6 V) to maintain a longlife measurement (larger than 10 h).

**Optical characterization**. The optical characterization of our devices is based on an inverted optical microscope (Olympus X71). An EMCCD (iXon 885) and a spectrometer (Andor Shamrock 750i) are used to capture the EL images and spectra, respectively. Detailed information is provided in Supplementary Note 2.

**Numerical simulations**. The far-field angular distributions and the emitted spectra are calculated based on the reciprocity theorem model described in Supplementary Note 5. We use the aperiodic Fourier modal method[51] to numerically compute the electric field in the tunnel junction barrier. The dielectric constants for Al are taken from ref. [52] and the Au data from ref. [53]. The refractive indexes of the glass substrate and alumina layer are set at 1.5 and 1.76, respectively. The modes of the patch antenna in Fig. 4g, h are calculated with the method described in ref. [54]. For that purpose, a vertically polarized (along $z$ direction) dipole source is inserted in the middle of the alumina layer along the $z$-axis and 30 nm off-center along the $x$-axis. More details can be found in Supplementary Note 5.

## Data availability
The data are available from the corresponding author upon reasonable request.

## Code availability
The codes are available from the authors upon reasonable request.

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

## Acknowledgements

This work was supported by a public grant by the French National Research Agency (ANR) project (ANR-15-CE24-0020). The authors thank the C2N nanofabrication platform at Paris-Sud University. C.Z. thanks the Ph.D financial support from the chair Safran-IOGS on Ultimate Photonics. J.-J.G. thanks the support of Institut Universitaire de France. The authors acknowledge enlighting discussions with Julien Gabelli. C.Z. acknowledges technical guidance from Jean-Rene Coudevylle.

## Author contributions

J.-J.G. conceived the idea, developed the theoretical model and supervised the project. C.Z. developed the fabrication procedures and fabricated the devices with help of A.-L.C. C.Z. built the experiment setup. C.Z. performed the experiments and analyzed the data with help from F.M. C.Z., C.S. and J.-P.H. carried out the simulations. J.-J.G. and C.Z. co-wrote the paper, with feedback from all co-authors.

## Competing interests

The authors declare no competing interests.
