## [Peer Review File · Nature Communications]

Reviewers' Comments:

Reviewer #1:

Remarks to the Author:

The authors did a decent job in answering the various comments of the referees. Overall, this is now a solid scientific paper, exhibiting both experimental results and an intuitive model to explain the data. In my view, this is now ready for publication in Nature communications.

Reviewer #3:

Remarks to the Author:

As I wrote in my previous report, I quite enjoyed reading this manuscript. I also now more clearly understand your motivations to generate plasmons and not photons! I am happy with your replies to my questions and thus recommend publication in Nature Communications.

Response letter to reviewers's remarks.

Reviewer #1 (Remarks to the Author):

The authors did a decent job in answering the various comments of the referees. Overall, this is now a solid scientific paper, exhibiting both experimental results and an intuitive model to explain the data. In my view, this is now ready for publication in Nature communications.

We thank the reviewer for his positive opinion.

Reviewer #3 (Remarks to the Author):

As I wrote in my previous report, I quite enjoyed reading this manuscript. I also now more clearly understand your motivations to generate plasmons and not photons! I am happy with your replies to my questions and thus recommend publication in Nature Communications.

We thank the reviewer for his comments and appreciation.